# Efficient Fatigue Modeling: Applying Operator Networks for Stress Intensity Factor Prediction and Analysis

## Abstract

Fatigue modeling is essential for material-related applications, including design, engineering, manufacturing, and maintenance. Central to fatigue modeling is the computation and analysis of stress intensity factors (SIFs), which model the crack-driving force and are influenced by factors such as geometry, load, crack shape, and crack size. Traditional methods are based on finite element analysis, which is computationally expensive. A common engineering practice is manually constructing handbook (surrogate) solutions, though these are limited when dealing with complex scenarios, such as intricate geometries. In this work, we reformulate SIF computation as an operator learning problem, leveraging recent advancements in data-driven operator networks to enable efficient and accurate predictions. Our results show that, when trained on a relatively small finite element dataset, operator networks — such as Deep Operator Networks (DeepONet) and Fourier Neural Operators (FNO) — achieve less than 5% relative error, significantly outperforming popular handbook solutions. We further demonstrate how these predictions can be integrated into crack growth simulations and used to calculate the probability of failure in small aircraft applications.

## 1 Introduction

In the mid-19th century, engineers noticed failures in bridges and railway components due to repeated loading. It quickly became clear that these failures were linked to the cyclic nature of the stress, often occurring without any prior warning. This phenomenon was identified as metal fatigue. A significant advancement in understanding metal fatigue is recognizing that structures often contain crack-like defects introduced during manufacturing. The central question in fatigue crack growth research is determining how long it takes for a crack to expand from an initial size to the maximum allowable size just before failure.

Understanding the period where the crack size grows requires the knowledge of a fatigue crack growth curve, as shown in Figure 1. The vertical axis is the crack growth rate, and the horizontal axis is the difference between the maximum and minimum stress intensity factor (SIF) during cyclic loading (Głuchowski & Sas, 2020), where SIF is denoted as $K$. SIF is a fundamental concept in fracture mechanics that describes the stress state near the tip of a crack and is the function of geometry and loading. It is used to predict the stress state near a crack tip and provide a failure criterion for materials (Tudose & Popa, 2007). In our work, stage 2 is of interest where crack growth rate $da/dN$ is of some power function of $\Delta K$, leading to a linear relation between the log of two quantities. Several attempts have been made to describe the crack growth rate curve using semi or wholly empirical formulae fitted to a data set. The most widely known is the Paris equation (Pugno et al., 2006):

$$\frac{da}{dN} = C(\Delta K)^m. \tag{1}$$

To obtain the constants $C$ and $m$, we need the SIF values $K$. Finite element (FE) analysis can effectively compute displacement fields from which SIFs can be calculated using M-integral (Banks-Sills et al., 2007). When converged, FE analysis can provide accurate solutions, but it requires substantial computational resources, especially within design iterations. As a result, researchers frequently turn to handbook solutions (Toribio et al., 2022), which act as efficient surrogate models and offer

Figure 1: Fatigue crack propagation stages. Figure is adapted from Tudose & Popa (2007)

a convenient way to estimate SIFs with reasonable accuracy. One most widely-used example is the manually-created Raju-Newman equations (Newman Jr & Raju, 1981) (Andersson) (Raju & Newman, 1979), which consists of high-order polynomial fits. These equations were developed from 3D FE analyses of cracks in finite elastic plates subjected to tension or bending loads. Raju-Newman equations can provide accurate SIF solutions for a wide range of crack geometries, including semi-elliptical surface cracks, quarter-elliptical corner cracks, and cracks near holes. However, these equations have limited application when it comes to complex geometry and crack shapes.

Machine learning (ML) offers a means to create surrogate models in a more flexible and accuracy way (Zhang et al., 2023). Such models can therefore be generalized to more complex geometries and boundary conditions due to their ability to express complex data. Merrell et al. (2024) used genetic programming based symbolic regression to learn equations that can predict SIF and improve the accuracy by 15-50% compared to Raju-Newman equations. Xia et al. (2022) introduce a SIF model for mode-I cracks in coal rock by training a convolutional neural network (CNN), that parameterizes the coal images and accurately predicts SIFs. Xu et al. (2022) focus on probabilistic failure risk assessment for an aero-engine disk. They conduct studies with Gaussian process (GP) regression, tree-structure models, and artificial neural networks (ANN). They show that the accuracy of SIFs can be improved by 5%–35%. Zhang et al. (2023) use ANN to predict mixed-mode SIFs of composites. The algorithm is trained on a dataset generated by combining the interaction integral and the extended FE method.

In this work, we introduce neural operators (Azizzadenesheli et al., 2024), a powerful new machine learning tool, for SIFs prediction with very high accuracy and generalizable to a wide range of geometries and crack shapes. SIFs are the function of geometry, crack shape, and the loading. The FE model defines the material geometry, crack shape, stresses, and displacement fields for loading. The displacement field from FE is then used by the M-integral method to evaluate SIFs along the crack front. This whole procedure can be seen as an operator learning problem where the combination of the FE model and M-integral act as an operator. In our dataset, material properties and loading conditions are fixed. The input to the operator is the function describing different geometries/crack shapes, and the output is the SIF along the crack front.

Our contributions are as follows:

1. Introducing FE SIF datasets for two different crack scenarios — a surface crack in a plate and a single corner crack at a shank hole in a plate — that provide different levels of problem complexity.

2. Applying three popular neural operators, including Deep Operator Network (DeepONet) (Lu et al., 2019), Fourier Neural Operator (FNO) (Li et al., 2020), and Proper Orthogonal Decomposition DeepONet (POD-DeepONet), (Lu et al., 2022) on the datasets, such that they act as surrogate models for predicting SIFs. We provide a comprehensive analysis of the model accuracy using the FE results as the benchmark.

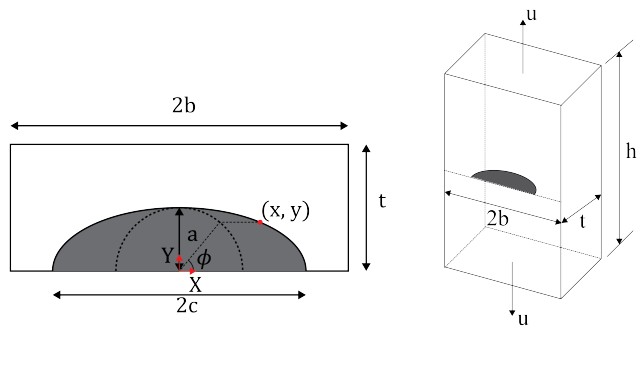

(a) Crack dimensions                    (b) Model geometry

Figure 2: (a) Crack parameters with $a$ being the crack depth, $2c$ being the surface crack length, and $\phi$ is defined by the angle to the inscribed circle projected to the ellipse. (b) Model geometry with plate height: h, plate width: 2b, and plate thickness: t. Tension loading is applied as a uniform displacement. The dark-shaded region represents the crack.

3. The results from operator networks are then used to simulate crack growth for a given geometry and initial crack shape. We show that the results from the operator networks can be used to investigate the probability of failure for a small aircraft against the number of flight hours.

## 2  DATASETS

### 2.1  SURFACE CRACK IN A PLATE

To generate the SIF dataset related to a surface crack in a plate, we employ high-fidelity FE models, representing semi-elliptical surface cracks under Mode I tension (Merrell et al., 2024). This dataset comprises several simulated results of $K$, each characterized by distinct plate and crack geometries shown in Figure 2. The FE models generate 2,956 different plate geometries and crack shapes represented using features $a/c$, $a/t$, $c/b$, and SIF values along the crack front represented using the parametric angle $\phi$. The resolution in $a/c$, $a/t$, $c/b$, and $\phi$ is 0.05, 0.05, 0.1, and 0.023 radians. The corresponding range of values in the features $a/c$, $a/t$, $c/b$, and $\phi$ is 0.2 to 2, 0.2 to 0.85, 0.01 to 0.3, and 0.057 to 3.088 radians, respectively. Out of 2956 different geometries and crack shapes, 2518 are used for training, and 438 test the accuracy of operator networks.

### 2.2  CORNER CRACK IN A PLATE

Quarter elliptic corner crack is another type of crack, and we discuss the SIF dataset for a corner crack at a shank hole in a plate. This scenario is generally considered more complex than a surface crack in a plain plate as it involves the intersection of multiple surfaces (the plate face, edge, and hole surface), creating a more complex crack front geometry. The shank hole introduces stress concentrations and alters the stress field around the crack, making the stress distribution more intricate, which affects the SIF distribution. FE results for this scenario were obtained from the Center for Aircraft Structural Life Extension (CASTLE) at the US Air Force Academy. The geometry used is shown in Figure 3. The FE models generate 24,519 different plate and crack geometries represented using features $W/r$, $a/c$, $a/t$, $r/t$, and SIF values along the crack front represented using the parametric angle $\phi$ (with a resolution of 0.023 radians). The corresponding range of values in the features $W/r$, $a/c$, $a/t$, $r/t$, and $\phi$ is 1.6 to 200, 0.1 to 10, 0.1 to 0.95, 0.5 to 1.5, and 0.052 to 1.52 radians, respectively. Out of the 24,519 different geometries and crack shapes, 18,389 are used for training, and 6,130 test the accuracy of operator networks.

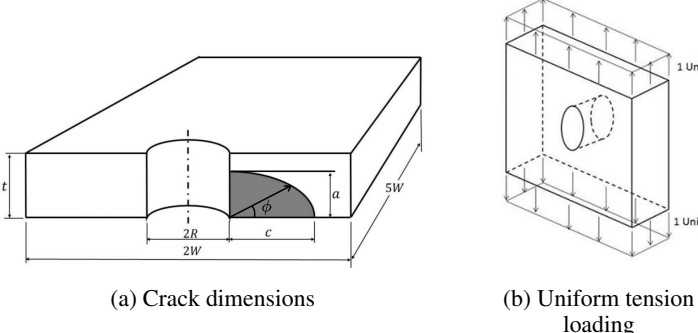

(a) Crack dimensions                    (b) Uniform tension
                                            loading

Figure 3: (a) Single crack scenario in a plate with dimensions $(2W, 5W, t)$ where the single crack of size $(a, c)$ satisfies $a < t$, $c + R \leq W$. The dark-shaded region represents the crack. (b) A basic load of unit 1 acts on the plate.

## 3  RESULTS

In this section, we present the results from the operator networks — DeepONet, POD-DeepONet and FNO, where they represent the mapping from geometry and crack shape to SIFs along the crack front. The details related to DeepONet, POD-DeepONet and FNO are explained in Appendix A.1, A.2 and A.3, respectively. We are using three metrics (described below) to compare the results against popular surrogate models — Raju-Newman equations for surface crack in a plate (Newman Jr & Raju, 1981) and Raju-Newman equations using Fawaz-Andersson solutions for corner crack at the hole (Raju & Newman, 1979) (Andersson).

1. Normalized Absolute Error (NAE) is defined for each SIF prediction, where $y_i$ is the SIF prediction and $\hat{y}_i$ is the FE SIF at the $\phi$ location for some geometry and crack shape,

$$E_i = \left| \frac{y_i - \hat{y}_i}{y_i} \right|. \tag{2}$$

2. Normalized Error (NE) is also defined for each SIF prediction, where $y_i$ is the SIF prediction and $\hat{y}_i$ is the FE SIF at the $\phi$ location for some geometry and crack shape,

$$E_i = \frac{y_i - \hat{y}_i}{y_i}. \tag{3}$$

3. Mean Normalized L2 Error (L2 error) is the average of all the normalized errors calculated at all the $\phi$ locations corresponding to different geometries and crack shapes, where vector $\mathbf{y_i}$ is the SIF prediction and vector $\hat{\mathbf{y}}_{\mathbf{i}}$ is the FE SIF at all $\phi$ locations for some geometry and crack shape,

$$E = \sum_{i=1}^{N} \frac{1}{N} \left( \frac{||\mathbf{y_i} - \hat{\mathbf{y}}_{\mathbf{i}}||_{\mathbf{2}}}{||\mathbf{y_i}||_{\mathbf{2}}} \right). \tag{4}$$

NE and NAE are commonly used metrics in solid mechanics (Daridon et al., 2020). We use NAE in the probabilistic estimates of different levels of errors and NE is used to estimate whether the model predictions are under or over the ground truth values. $L_2$ error is the most commonly used metric in ML studies (Li et al., 2020) because it allows the comparison of errors across different datasets or models and the standardization makes it easier to assess relative performance.

### 3.1  SURFACE CRACK IN A PLATE

Raju-Newman equations for surface crack (Newman Jr & Raju, 1981) are empirical equations developed using the 3D FE dataset. They are mechanics-driven equations and have been validated against experimental data. There have been several updates to these equations, but the application is still limited regarding complex geometries (Bocher et al., 2018). Figure 8 shows the complementary

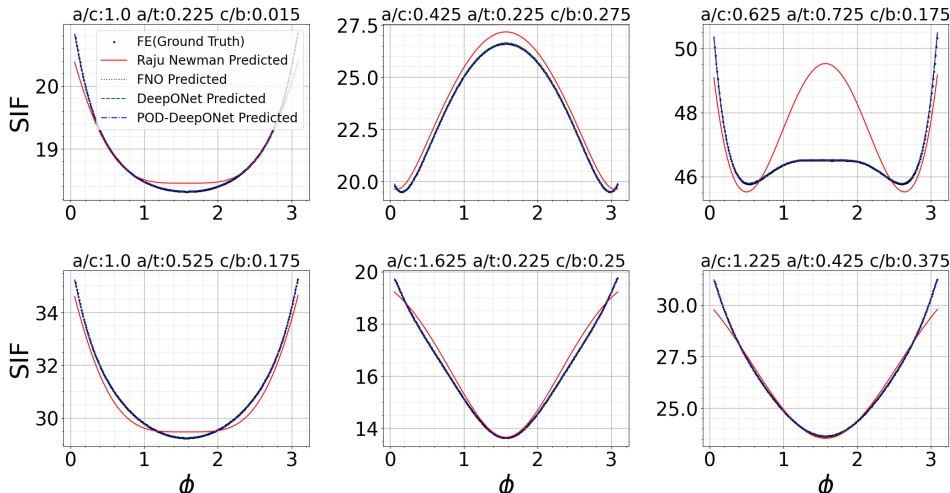

Figure 4: Comparison of the surface crack SIF predictions.

cumulative density function (1-CDF) of the testing NAEs for Raju-Newman equations, DeepONet, POD-DeepONet, and FNO trained on the surface crack in a plate dataset. From the results, we can see that all three operator networks provide a significant improvement over the Raju-Newman equations. All the errors for operator networks are under 3.5%, and the probability of NAE being greater than 1% is 1/1000. However, for Raju-Newman equations, the error can be as high as 17%, and the probability of NAE being greater than 1% is around 1/4.

Figure 4 compares the predictions from all models for different geometries and crack shapes. We can see that predictions from Raju-Newman equations are inconsistent. Operator networks, however, predict with good accuracy for all the examples. Figure 5a compares the predictions along the crack front ($\phi$) from all three operator networks when $a/c < 1$ (i.e., the crack is wider than its depth). We can see that POD-DeepONet has the largest errors, which are present away from the free surface where $\phi \approx 0$ or $\phi \approx \pi$. Figure 5b shows the results when $a/c > 1$ (i.e., the crack is deeper than its width). In this case, we notice low errors for all $\phi$ values. This shows that the errors depend largely on the geometry and the crack shape. This is expected because the complexity in SIF values is influenced by geometry and crack shape.

## 3.2 CORNER CRACK IN A PLATE

Raju-Newman equations using Fawaz-Andersson solutions for corner crack at the hole (Raju & Newman, 1979) (Andersson) are also empirical equations discovered from the FE dataset with limited application in terms of geometry and crack shape. More specifically, they are limited to — $0.2 \leq a/c \leq 2$, $a/t \leq 0.8$, $0.5 \leq r/t \leq 2$, and $((r + c)/b) < 0.5$. Developing these equations is also costly as it requires engineers with domain-specific knowledge and takes a significant amount of time (months/years to develop and validate the equations). Figure 9 shows 1-CDF of the testing NAEs for Raju-Newman equations, DeepONet, POD-DeepONet, and FNO trained on the corner crack in a plate dataset. From the results, we can see that all three operator networks provide a significant improvement over the Raju-Newman equations. The errors for operator networks are under 5.5%, and the probability of NAE being greater than 1% for DeepONet, POD-DeepONet, and FNO is around 9/10000, 2/10000, and 1/1000, respectively. However, for Raju-Newman equations, the error can be as high as 17.5%, and the probability of NAE being greater than 1% is around 1/2.

Figure 6 compares the predictions from all models for different geometries and crack shapes. We can see that Raju-Newman equations are again inconsistent (especially when the SIF distribution is complex). Operator networks, however, predict with good accuracy. Figures 7a compare the predictions along the crack front ($\phi$) from the operator networks when $a/c < 1$ (i.e., the crack is wider than its depth). We can see that FNO has the largest errors, which are present close to the surface where $\phi \approx 0$. Figure 7b shows the results when $a/c > 1$ (i.e., the crack is deeper than its width). In this case, we also notice relatively large errors for $\phi \approx 0$ values. This shows an interesting

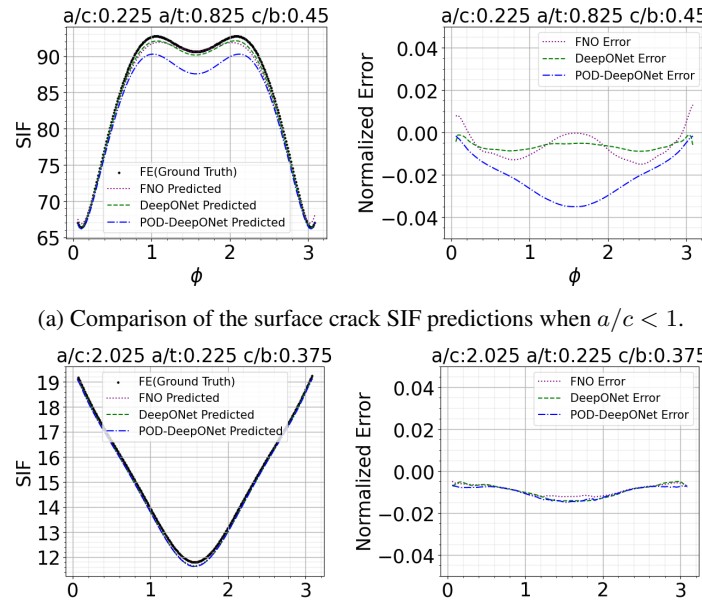

(a) Comparison of the surface crack SIF predictions when $a/c < 1$.

(b) Comparison of the surface crack SIF predictions when $a/c > 1$.

Figure 5: Comparison of the surface crack SIF predictions.

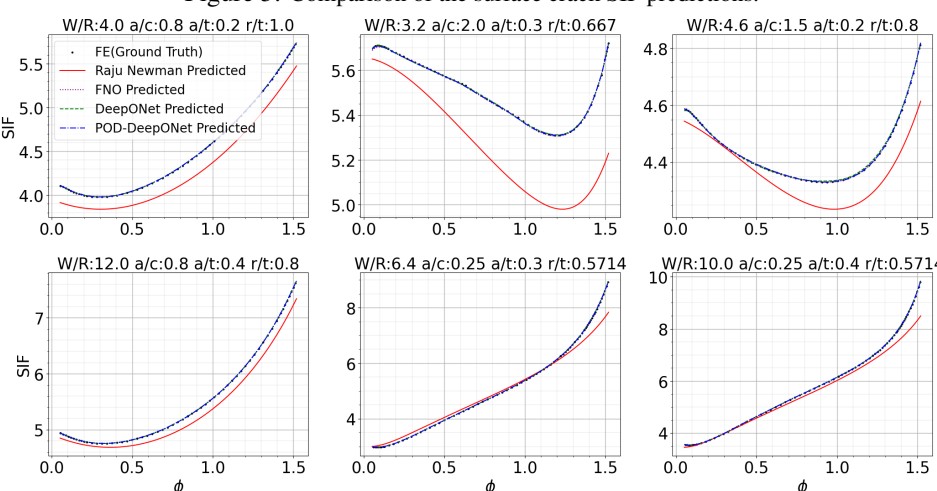

Figure 6: Comparison of the corner crack SIF predictions.

trend where the errors depend on the geometry, crack shape, and position along the crack front. This is expected because the complexity in SIF values is influenced by these factors.

Table 1 shows the $L_2$ error for all the dataset scenarios and the models. Operator networks have very similar accuracy for both datasets and compared to Raju-Newman equations, they are several orders of magnitude better.

Table 1: Mean normalized $L_2$ error on the test dataset for all the models.

|  | DeepONet | POD DeepONet | FNO | Raju-Newman |
|---|---|---|---|---|
| **Surface Crack in Plate** | 0.000776 | 0.000817 | 0.000695 | 0.023611 |
| **Corner Crack in Plate** | 0.000755 | 0.000530 | 0.000627 | 0.036049 |

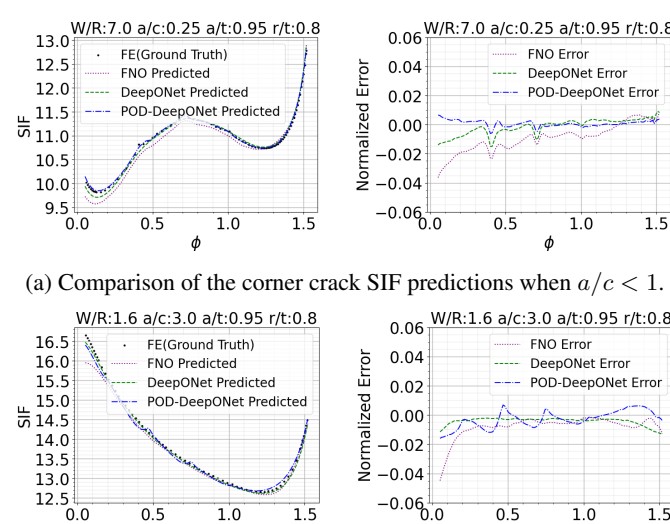

(a) Comparison of the corner crack SIF predictions when $a/c < 1$.

(b) Comparison of the corner crack SIF predictions when $a/c > 1$.

Figure 7: Comparison of the corner crack SIF predictions.

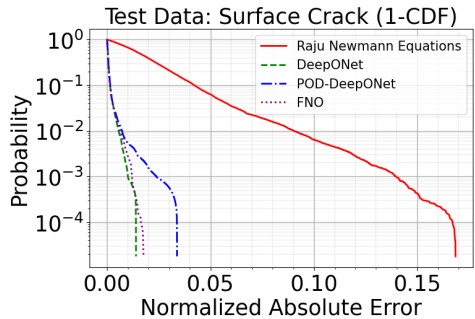

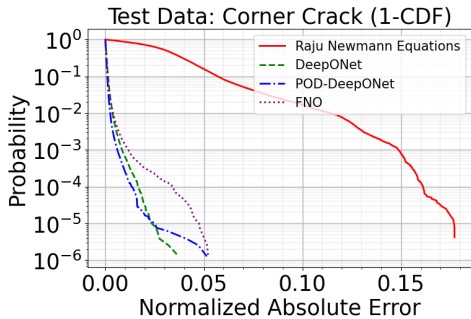

Figure 8: 1-CDF of the errors on the surface crack test dataset.

Figure 9: 1-CDF of the errors on the corner crack test dataset.

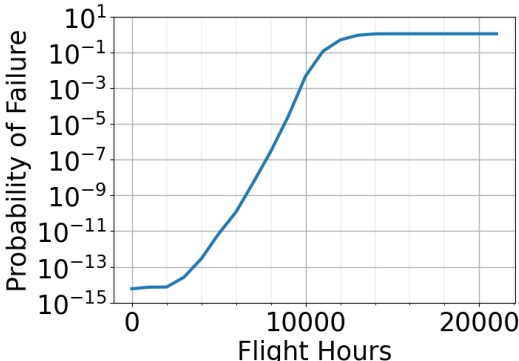

Figure 10: Probability of failure calculated using SIF values from DeepONet.

## 4    APPLICATIONS TO FATIGUE MODELING

SIF is a fundamental concept in fracture mechanics that characterizes the stress state near the tip of a crack. It plays a crucial role in analyzing crack behavior and predicting material failure. In this section, we show how the operator networks can be used to simulate the crack growth and perform damage tolerance modeling (Millwater et al., 2019). Crack growth simulation using SIFs involves a few steps. SIF is calculated for the crack geometry and loading conditions, which characterize the stress state near the crack tip. This is where the operator networks are used. Empirical crack growth laws like Paris' law (shown in Equation 1) relate the crack growth rate to the SIF range. The crack is grown incrementally based on the calculated growth rate, and the crack geometry is updated after each increment. As the crack grows, the SIFs need to be recalculated for the new crack geometry.

For fatigue crack growth, the process is repeated several times throughout the load cycles. Growth is simulated until a critical crack size is reached, indicating failure. The whole process starts with the SIF values, and if the predictions are not accurate, the resulting crack growth simulation will accumulate large errors. Fast inference from operator networks is another key factor that speeds up the simulation. Throughout the crack growth simulation, SIF values are required hundreds of times. This makes using FE methods prohibitive as they can take significantly longer to calculate SIF values.

A corner crack in a plate is the more complicated case, and we are using this to demonstrate the crack growth simulation. For example, consider an initial crack with lengths 0.0125 and 0.0125. This crack is present in the plate with full width ($2W$) 3.75, thickness ($t$) 0.125, and hole diameter ($2r$) 0.25. Figure 12 shows the geometry for this example alongside the SIF predictions from the operator networks against the FE model for this initial crack. We can see that the predictions from all three models are close to the FE results. Operator networks can then be used in hypergrow (Ocampo et al., 2020) to simulate the crack growth. As the predictions from the three operator networks are very similar, the resulting crack growth results will also be very similar. Figure 13 shows the crack progression for DeepONet and POD-DeepONet with the number of load cycles. We can see that the results from both these models are very similar. As loading cycles accumulate, crack lengths grow in both directions and around 70000 cycles, the crack reaches the thickness of the plate. This is where it changes from a corner crack to a through crack (Taylor et al., 2005). Simulating the results further, we can see the crack growing and it will eventually lead to failure. As FE takes a really long time to calculate SIFs, it is impractical to get results for every cycle. The only thing feasible in this case is to get FE runs for a few discrete cases and compare the SIF predictions from the operator networks against the FE SIFs. Figures 11 shows different crack geometries and the SIF predictions from operator networks alongside FE results. We can see that for all the cases, the predictions are very close to the FE results with an absolute error of less than 1%.

SIFs from the operator networks can also be used in SMART-DT (Small Aircraft Risk Technology - Damage Tolerance), which is a probabilistic damage tolerance analysis (PDTA) software developed with application to small aircraft risk assessment (Millwater et al., 2019). Damage tolerance analysis involves generating a deterministic crack growth analysis and finding the time at which the crack can be found by inspection with 95% confidence given a non-destructive inspection technique and the time at which the crack grows to critical size. The time to first inspection and inspection interval is set as the time when the crack is detectable and half of the time between when the crack becomes detectable and the time to failure. PDTA calculates the single flight probability of failure — the probability that a component with existing damage fractures on a flight has survived all prior flights. PDTA allows for uncertainties in usage, material properties, crack growth rate, and initial damage size to be incorporated into the risk analysis, but the computational expense increases considerably when uncertainty in crack growth rate is included, as this requires performing many crack growth analyses. The probability of failure (PoF) in Figure 10 is evaluated from 3900 crack growth analyses using DeepONet SIF model in 31.962 seconds. The PoF calculated using PDTA provides a forecast of when fatigue cracks will begin to grow large enough to fracture. Planning maintenance actions to inspect and repair such that PoF remains below an acceptably low risk threshold, $10^{-7}$ as noted by Lincoln (1985), ensures that failures due to fatigue cracking will be extremely rare events. In addition to being used as a tool for scheduling maintenance proactively, PDTA can also be used to estimate the economic life of aircraft and model the effects of changing the aircraft usage.

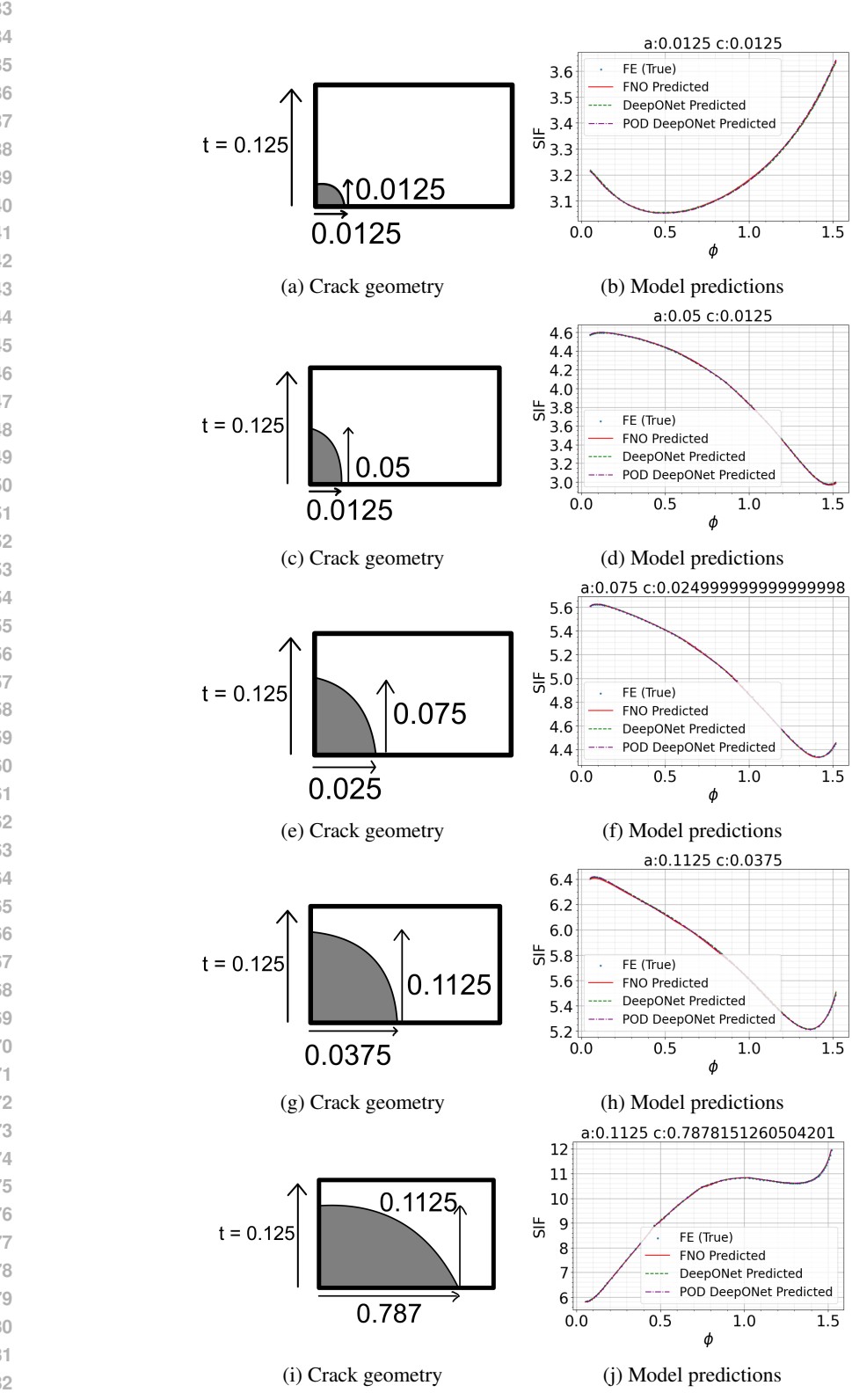

Figure 11: Left column shows single crack in a plate where the dark-shaded region represents the crack with right column showing the corresponding predictions from operator networks alongside FE results.

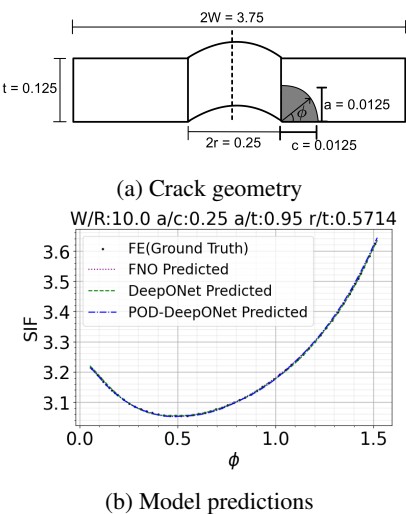

(a) Crack geometry

(b) Model predictions

Figure 12: (a) Single crack scenario in a plate where the dark-shaded region represents the crack. (b) Predictions from operator networks plotted alongside FE results.

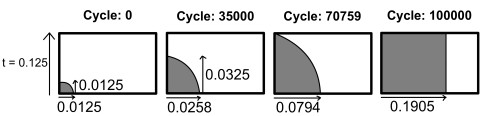

(a) Crack growth simulation using DeepONet where loading cycles increase from left to right

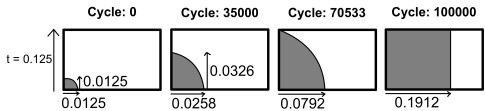

(b) Crack growth simulation using POD-DeepONet where loading cycles increase from left to right

Figure 13: Crack progression in a plate using SIF values where the dark-shaded region represents the crack.

## 5 CONCLUSIONS

For complex crack growth scenarios, the operator networks can predict SIF values accurately with errors less than 5%. This is several orders of magnitude better than the widely used handbook solutions like Raju-Newman equations, and at par with the industry standard FE models. Operator networks can be trained effectively on a relatively small dataset size and once trained, the predictions can be made very quickly. For fatigue crack growth simulation, SIF values are required several times. FE methods are not suitable as they take a significant amount of time to calculate SIFs given geometry and crack shape. Operator networks, can be used to make predictions quickly, and in this work, we were able to simulate crack growth over 100000 load cycles within 0.5 seconds with an accuracy of around 99%. Probability of failure can also be computed using multiple crack growth analysis and using operator networks this can be done in a few seconds.

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

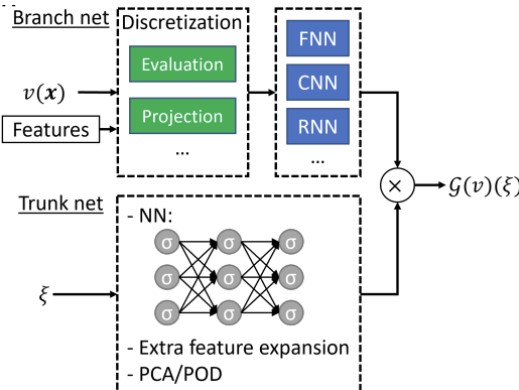

Figure 14: Architecture of DeepONet and POD DeepONet. Figure is taken from Lu et al. (2022).

# A APPENDIX

## A.1 DEEPONET

DeepONet is based on the universal approximation theorem for operators, which states that for any continuous nonlinear operator $G$ and any $\epsilon > 0$, there exists a DeepONet that can approximate $G$ with an error less than $\epsilon$. DeepONet, shown in Figure 14 is a neural network architecture designed to learn nonlinear operators, based on the universal approximation theorem for operators. It consists of two sub-networks — a branch network $B(u)$ that encodes the input function $u$ at $m$ fixed locations $x_i{}_{i=0}^{m-1}$, and a trunk network $T(y)$ that encodes the output locations $y$. The DeepONet output is given by:

$$G(u)(y) = \sum_{i=0}^{p-1} b_i(u) \cdot t_i(y) \tag{5}$$

where $b_i(u)$ and $t_i(y)$ are the outputs of the branch and trunk networks, respectively, and $p$ is the width of the last layer in both networks. This architecture allows DeepONet to efficiently learn complex operator mappings $G : \mathcal{U} \to \mathcal{V}$ between function spaces, with applications in solving differential equations and modeling dynamical systems. DeepONet is a high-level framework that does not restrict branch and trunk networks to any specific architecture. $y$ is usually low dimensional which makes standard fully connected neural nets a good choice for trunk net, but the choice of branch net depends on the type of input functions $u$.

## A.2 POD-DEEPONET

POD-DeepONet, shown in Figure 14 is an enhanced version of the original DeepONet architecture that incorporates Proper Orthogonal Decomposition (POD) to improve efficiency and accuracy in learning nonlinear operators. In vanilla DeepONet basis of the output function are learned using the trunk net. In the POD-DeepONet, these basis are obtained by performing POD on the training data (after first removing the mean). Then, these POD basis are used alongside the branch net (that learns the coefficients of the POD basis) to get the output. This can be formally written as:

$$G(u)(y) = \sum_{i=0}^{p-1} b_i(u) \cdot \phi_i(y) + \phi_0(y) \tag{6}$$

where $\phi_0(y)$ is the mean function computed from the training data, where $b_i(u)$ and $phi_i(y)$ are the outputs of the branch and the POD basis, respectively, and $p$ is the number of basis for the problem.

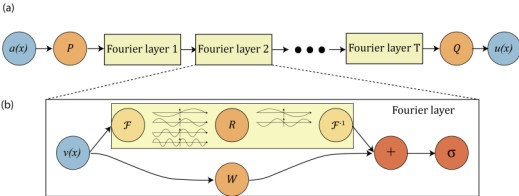

Figure 15: Architecture of FNO. Figure is taken from Li et al. (2020).

## A.3 FNO

Fourier Neural Operators (FNOs), shown in Figure 15 are designed to learn mappings between infinite-dimensional function spaces, typically of the form $G : \mathcal{A} \to \sqcap$ where $\mathcal{A}$ and $\mathcal{U}$ are Banach spaces. The key innovation of FNOs is the Fourier layer, which performs convolutions in the spectral domain. Given an input function $a(x)$, with aim to predict $u(x)$, the FNO architecture has the following key features:

1. **Lifting Layer:** A standard neural net $P$ lifts the input to a high dimensional space: $v_0(x) = P(x, a(x))$.

2. **Fourier Layers:** A series of Fourier and inverse Fourier transforms are then applied (Li et al., 2020): $v_l(x) = \sigma(W v_{l-1}(x) + (\mathcal{F}^{-1}(R_l \mathcal{F}(v_{l-1})))(x))$.

3. **Projection Layer:** Finally, another neural net $Q$ projects the output from fourier layers back to the low dimensional space: $u(x) = Q(x, v_L(x))$.

This formulation allows FNOs to handle a wide range of problems in scientific computing, offering a powerful alternative to traditional numerical methods, especially for complex, high-dimensional systems.

