# OpenReview forum: "Efficient Fatigue Modeling: Applying Operator Networks for Stress Intensity Factor Prediction and Analysis"
_ICLR.cc/2025/Conference — Submitted to ICLR 2025_

### Official Review · Reviewer_hyG6 · 2024-10-30

**Soundness:** 2
**Presentation:** 3
**Contribution:** 3
**Rating:** 5
**Confidence:** 4

**Summary:**

The paper presents the application of neural operators to enhance fatigue Modeling while maintaining a high level of accuracy. A dataset featuring diverse geometries and crack types is created through finite element (FE) simulations. This data is utilized to develop operators capable of predicting the Stress Intensity Factor (SIF) in near real-time. The effectiveness of these operators is demonstrated in crack growth simulations, achieving significant acceleration compared to traditional FE methods. Additionally, the approach is shown to be more accurate than the handbook solutions commonly employed in the industry.

**Strengths:**

The paper is well written. Accelerating FE simulations is an important problem in the field of numerical simulations. The need for faster inference is clearly articulated. Neural operators naturally lend themselves to the problem. However, their application is not straightforward.
The paper presents a very interesting application of neural operators for a practical problem in the industry. The idea of using neural operators for mitigating the repetitive bottlenecks in fatigue modelling looks novel. The methodology is reasonably clear barring few details which seem to be skipped. Claims in the paper are well supported by the results identifying the benefits over conventional methods.

**Weaknesses:**

A thorough exploration of prior work on fatigue modeling using PINNs, neural networks and machine learning approaches, pointing out their limitations would strengthen the case of this work.

While this is an interesting engineering application of neural operators, it seems to use only vanilla neural operator frameworks. The details of the modeling effort - architecture used, learning strategies tweaked for this problem, challenges faced during training, hyperparameters selection, loss curves are all missing from the paper. It is very difficult to evaluate the contribution without these details.

The results look good against conventional handbook methods, but they should also be compared against other ML approaches - PINNs, neural networks, ML methods.

Is there any validation for probability of failure with real life data? If yes, that should be added as well.

The ground truth in Figures 4,5,6 is very hard to see. Suggest changing the symbol/colour/font size to make it easy to comprehend.

**Questions:**

Is it possible to use neural operators for capturing entire transient fatigue modeling? Instead of switching between crack growth calculation and SFI prediction. What would be the challenges in this?

What challenges do you envisage for operator learning when loading conditions and material properties change?

What about performance on new out-of-distribution scenarios? How about using physics equations as a constraint while building these operators? As an example, can something similar be done within neural operator framework - https://www.sciencedirect.com/science/article/abs/pii/S0167844224004671?via%3Dihub

---

> ### Author Response · Authors · 2024-11-19
>
> We would like to thank the reviewer for the comments. Responses to the comments **C** are marked **R**:
>
> ---
>
> ***C1***: A thorough exploration of prior work on fatigue modeling using PINNs, neural networks and machine learning approaches, pointing out their limitations would strengthen the case of this work.
>
> ***R1***: We will add a detailed comparison against ANN, Random Forest Regression (RFR) and SVM in the paper. Comparison against PINNs is not possible because we don't have any physics equations constraining the SIF computation. The summary of the L2 errors are shown below:
> | |DeepONet|POD-DeepONet|FNO|Raju-Newman Equations|ANN|RFR|SVM|
> |-|--------------|----------------------|------|---------------------------------|-------|------|------|
> | Surface Crack | 0.000776|0.000817|0.000695|0.023611|0.000670|0.037400|0.005107|
> | Corner Crack |0.000755| 0.000530| 0.000627| 0.036049| 0.001018|0.039508|0.136749|
>
> From this we can see that using operator networks are at par with ANN for the surface crack, while we get atleast an order of magnitude improvement over other ML algorithms for more complex corner crack dataset.
>
> ---
>
> ***C2***: *While this is an interesting engineering application of neural operators, it seems to use only vanilla neural operator frameworks. The details of the modeling effort - architecture used, learning strategies tweaked for this problem, challenges faced during training, hyperparameters selection, loss curves are all missing from the paper. It is very difficult to evaluate the contribution without these details.*
>
> ***R2***: We thank the reviewer. We will add the details in the appendix.
>
> ---
>
> ***C3***: *The results look good against conventional handbook methods, but they should also be compared against other ML approaches - PINNs, neural networks, ML methods.*
>
> ***R3***: Please see the response ***R1*** for this comment.
>
> ---
>
> ***C4***: *Is there any validation for probability of failure with real life data? If yes, that should be added as well.*
>
> ***R4***: Validating real life probability of failure would require millions of experiments/inspections of the aircrafts. This will take significant amount of time/resources and is outside the scope of this work.
>
> ---
>
> ***C5***: *Is it possible to use neural operators for capturing entire transient fatigue modeling? Instead of switching between crack growth calculation and SFI prediction. What would be the challenges in this?*
>
> ***R5***: Once we have accurate SIF values, the crack growth simulation is performed by solving a relatively simple differential equation. We have methods like Runge-Kutta that can do this very effectively. Such an integration approach is important because the crack growth depends on the sequence of cycles applied. For example, it’s not the case that constant amplitude cycles are applied. Instead, any random sequence of cycle amplitudes can be applied, which is an intractable problem for ML, considering that millions of cycles are often applied. This will lead to lower accuracy and slower training/prediction times.
>
> ---
>
> ***C6***: *What challenges do you envisage for operator learning when loading conditions and material properties change?*
>
> ***R6***: We thank the reviewer for the attention to detail. We made a typo in the manuscript, where we mentioned that SIF depends on the material property. We will correct it to reflect that SIF only depends on the geometry and loading conditions (and not on material properties). Regarding loading conditions, it does not change the problem complexity, hence will not pose any significant challenge for the operator networks. In accordance with linear elastic fracture mechanics, multiple loading conditions would simply be superpositions on the tension loading. For reference, please check out Section 2.6 of the book “[Fracture Mechanics](https://www.taylorfrancis.com/books/mono/10.1201/9781482265583/fracture-mechanics-jan-zuidema-michael-janssen-russell-wanhill) by *Michael Janssen, Jan Zuidema, Russell Wanhill*”.
>
> ---
>
> ***C7***: *What about performance on new out-of-distribution scenarios? How about using physics equations as a constraint while building these operators? As an example, can something similar be done within neural operator framework - https://www.sciencedirect.com/science/article/abs/pii/S0167844224004671?via%3Dihub*
>
> ***R7***: Sampling out of distribution is not feasible in our case because the dataset is generated practically and we can’t sample any parameters like crack shape and manipulate the dataset to get out of distribution examples. This will lead to cases that are not practical and will offer nothing. There are no physics equations describing the problem, hence we can not constrain the learning. The paper link provided by the reviewer is from a different (unrelated) problem and is not applicable for our work.

---

> > ### Comment · Reviewer_hyG6 · 2024-11-28
> >
> > Thank you authors for your comments.
> > For comment C2, would like to see those details related to the training. I don't see them in the appendix yet. Add them in the response here if you are not able to edit the appendix now. Difficult to properly evaluate unless those are added. Thanks.

---

### Official Review · Reviewer_TxYD · 2024-10-31

**Soundness:** 2
**Presentation:** 3
**Contribution:** 2
**Rating:** 5
**Confidence:** 3

**Summary:**

The authors use operator networks to predict SIFs with high efficiency and accuracy and can be generalizable to a wide range of geometries and crack shapes. The combination of the FE model and M-integral acts as an operator. The capabilities of models are demonstrated by integrating the learned operator into crack growth simulations and calculating the probability of failure in small aircraft applications.

**Strengths:**

1. They reformulate the SIF computation as an operator learning problem.

2. The SIFs are predicted with high efficiency and accuracy, which are validated on datasets of surface and corner cracks in plates.

3. The framework can be integrated into crack growth simulations and used to calculate the probability of failure in small aircraft applications.

**Weaknesses:**

1. The methodological innovation is not prominently highlighted. It seems to be a combination of existing approaches such as FE modeling, operator networks.

2. The details of the method may require further elaboration, such as the process of neural network training and the setting of hyperparameters. Additionally, information on how the training and test datasets were divided, and which crack geometries were used for training versus testing, should be specified.

3. The framework lacks more demonstrative experimental data to verify its feasibility. The dataset, trained through FE models, may have deviations in experimental scenarios compared to the FE models (such as in material constitutive models, geometry, loading conditions, etc.).

**Questions:**

1. Please further elucidate the methodological innovation. It should not merely be a concatenation of existing methods.

2. Please add the details of the method, such as the process of neural network training and the setting of hyperparameters. Additionally, information on how the training and test datasets were divided, and which crack geometries were used for training versus testing, should be specified to better evaluate the generalizability of models.

3. Provide examples of the best, worst, and median performance of the proposed machine learning model. Show the prediction of the closest point in the dataset of each of those examples, so that readers understand the quality of the machine learning model.

4. The operator implicitly encapsulates material constitutive relations, loading conditions, etc., within the finite element model. When the method is applied to real-world engineering scenarios with uncertainties compared to the training environment, a discussion on the model’s applicability should be expanded.

---

> ### Author Response · Authors · 2024-11-19
>
> We would like to thank the reviewer for the comments. Responses to the comments **C** are marked **R**:
>
> ---
>
> ***C1***: *The methodological innovation is not prominently highlighted. It seems to be a combination of existing approaches such as FE modeling, operator networks. Please further elucidate the methodological innovation. It should not merely be a concatenation of existing methods.*
>
> ***R1***: This paper is an applications paper and focuses on applying neural operator to the practical problem of fatigue modeling. We show that using operator networks, the results can be further improved and predictions can be made very quickly, which results in very fast crack growth simulation. ICLR encourages papers showing the application to complex real-life problems and we aim to target that with this paper.
>
> ---
>
> **C2**: *The details of the method may require further elaboration, such as the process of neural network training and the setting of hyperparameters. Additionally, information on how the training and test datasets were divided, and which crack geometries were used for training versus testing, should be specified.*
>
> ***R2***: We thank the reviewer for the comment. We will add this in the appendix.
>
> ---
>
> ***C3***: *The framework lacks more demonstrative experimental data to verify its feasibility. The dataset, trained through FE models, may have deviations in experimental scenarios compared to the FE models (such as in material constitutive models, geometry, loading conditions, etc.).*
>
> ***R3***: SIFs cannot be measured experimentally. They can be computed but not measured. The scope of this study is to demonstrate accurate SIF predictions. SIFs are a function of geometry and loading conditions. Indeed, both will have associated uncertainties. To test this, a significantly expanded dataset to characterize the propagation of uncertainty through the FE model for SIF would need to be completed, then a Bayesian or similar training approach (like deep ensemble) would be undertaken. While interesting, this is far outside the scope of paper.
>
> ---
>
> ***C4***: *Provide examples of the best, worst, and median performance of the proposed machine learning model. Show the prediction of the closest point in the dataset of each of those examples, so that readers understand the quality of the machine learning model.*
>
> ***R4***: We thank the reviewer of the comment. We will add this to the paper.
>
> ---
>
> ***C5***: *The operator implicitly encapsulates material constitutive relations, loading conditions, etc., within the finite element model. When the method is applied to real-world engineering scenarios with uncertainties compared to the training environment, a discussion on the model’s applicability should be expanded.*
>
> ***R5***: Please check ***R3*** for this comment as well. We briefly discuss uncertainty analysis using SMART-DT and plot the probability of failure, but a more detailed analysis is for the future work.

---

### Official Review · Reviewer_mZNZ · 2024-11-02

**Soundness:** 2
**Presentation:** 2
**Contribution:** 2
**Rating:** 3
**Confidence:** 5

**Summary:**

The work compares the performance of three operator networks in predicting the stress intensity factors in model structures. Based on two sets of finite element datasets on representative plate and holed-specimen geometries, the operator neural network approach outperforms approximate textbook solutions solved from the Newman equations, with very high computational efficiency (0.1 M loading cycles within 0.5 s). The workflow to integrate the calculated stress intensity factors into fatigue crack growth is discussed.

**Strengths:**

Finite element simulation datasets are created, which allow the assessment of operator neural networks in predicting the stress intensity factor solutions from the finite-geometry linear elastic problems under specific loading conditions. The data-driven computation of stress intensity factors is orders of magnitudes faster than the finite element solvers, which very good accuracy for the different geometries and crack shapes (for plates and holed specimens). The work shows the potential of applying these methods into fatigue performance assessment.

**Weaknesses:**

The loading conditions is limited to uniform tension. The authors are suggested to explore the performance of predictions under more complex loading conditions such as non-uniform tension and a combination of tension and shear. Similar, it is not clear how the model trained using the datasets constructed in the current work generalizes to specimens and cracks with very different geometries and shapes such as a plate with varying thickness, and a solid with irregular or 3D crack geometries, which are essential if one considers practical applications of these ideas. The detailed parameters and settings of the operator neural network models should be given in the appendix.

**Questions:**

How does the model trained for the plates apply to the holed specimens, and vice versa? The authors are suggested to introduce quantitative performance metrics when applying the model trained on one geometry to the other, compared to when it's trained on the specific geometry. How do the three operator neural networks compare in terms of computational costs and speeds?

---

> ### Author Response · Authors · 2024-11-19
>
> We would like to thank the reviewer for the comments. Responses to the comments **C** are marked **R**:
>
> ---
>
> ***C1***: *The loading conditions is limited to uniform tension. The authors are suggested to explore the performance of predictions under more complex loading conditions such as non-uniform tension and a combination of tension and shear.*
>
> ***R1***: Adding bending or bearing loading to tension does not change the problem complexity. In accordance with linear elastic fracture mechanics, these additional loading conditions would simply be superpositions on the tension loading. For reference, please check out Section 2.6 of the book “[Fracture Mechanics](https://www.taylorfrancis.com/books/mono/10.1201/9781482265583/fracture-mechanics-jan-zuidema-michael-janssen-russell-wanhill) by *Michael Janssen, Jan Zuidema, Russell Wanhill*”.
>
> ---
>
> ***C2***: *It is not clear how the model trained using the datasets constructed in the current work generalizes to specimens and cracks with very different geometries and shapes such as a plate with varying thickness, and a solid with irregular or 3D crack geometries, which are essential if one considers practical applications of these ideas.*
>
> ***R2***: From a fracture mechanics expert perspective, such a generalization would never be possible because “very different geometries and shapes” would require new models due to requirement of adding new features. Additionally, it is never the case in practice “irregular or 3D crack geometries” are considered for practical application. Instead, surrogate models of idealizations of more complex scenarios are used in every standard procedure for estimating fatigue crack growth and damage tolerance in practice, e.g., SIF model use in NASGRO and AFGRO.
>
> ---
>
> ***C3***: *The detailed parameters and settings of the operator neural network models should be given in the appendix.*
>
> ***R3***: We thank the reviewer for the comment. We will add this to our paper.
>
> ---
>
> ***C4***: *How does the model trained for the plates apply to the holed specimens, and vice versa? The authors are suggested to introduce quantitative performance metrics when applying the model trained on one geometry to the other, compared to when it's trained on the specific geometry.*
>
> ***R4***: The two datasets have different features. So, training on one and testing on the other is not sensical.
>
> ---
>
> ***C5***: *How do the three operator neural networks compare in terms of computational costs and speeds?*
>
> ***R5***: We thank the review for the suggestion. We will add this in the appendix. In summary, POD-DeepONets trained the fastest, followed by DeepONets and FNO was the slowest. Computational cost follows the order where FNO is most intensive, followed by DeepONet and POD-DeepONet.

---

> > ### Comment · Reviewer_mZNZ · 2024-11-30
> > **Comments on the authors reply**
> >
> > From the replies R1, R2, and R4, it is difficult to see what is the advantage of the current approach in addressing real-world fatigue design problems that cannot be done using existing non-machine learning methods. Maybe the authors can add a showcase to highlight this.

---

### Official Review · Reviewer_J9iU · 2024-11-08

**Soundness:** 2
**Presentation:** 1
**Contribution:** 2
**Rating:** 3
**Confidence:** 4

**Summary:**

This paper evaluates the effectiveness of neural operators for the problem of fatigue modeling. Three neural operator learning methods are used, namely FNO, DeepONet, and POD-DeepONet for predicting crack growth on simulated datasets. Results show comparison of neural operator methods with a numerical method considered as ground-truth.

**Strengths:**

1. Studies an interesting real-world problem with application to a new area of engineering
2. Provides citations to relevant literature in machine learning for fatigue modeling

**Weaknesses:**

1. Weak comparison with baselines. While the paper mentions several previous works in using machine learning methods such as ANNs in fatigue modeling, none of them have been compared. This makes it hard to evaluate the importance of operator learning methods compared to previous works.
2. It is not clear where the novelty of this paper lies, since they explore applications of known neural operator methods on a new dataset. It will be useful to explicitly state the contributions of this work.
3. Limited complexity of the dataset. The visualizations suggest that the problem involves predicting a single 1D variable that is smoothly varying. More details can be provided on the complexity of tasks considered, and a possible categorization of data samples based on the level of complexity.

**Questions:**

See weaknesses above.

---

> ### Author Response · Authors · 2024-11-19
>
> We would like to thank the reviewer for the comments. Responses to the comments **C** are marked **R**:
>
> ---
>
> **C1:** *Weak comparison with baselines. While the paper mentions several previous works in using machine learning methods such as ANNs in fatigue modeling, none of them have been compared. This makes it hard to evaluate the importance of operator learning methods compared to previous works.*
>
> **R1:** We will add a detailed comparison against ANN, Random Forest Regression (RFR) and SVM in the paper. The summary of the L2 errors are shown below:
> | |DeepONet|POD-DeepONet|FNO|Raju-Newman Equations|ANN|RFR|SVM|
> |-|--------------|----------------------|------|---------------------------------|-------|------|------|
> | Surface Crack | 0.000776|0.000817|0.000695|0.023611|0.000670|0.037400|0.005107|
> | Corner Crack |0.000755| 0.000530| 0.000627| 0.036049| 0.001018|0.039508|0.136749|
>
> From this we can see that using operator networks are at par with ANN for the surface crack, while we get atleast an order of magnitude improvement over other ML algorithms for more complex corner crack dataset.
>
> ---
>
> **C2:** *It is not clear where the novelty of this paper lies, since they explore applications of known neural operator methods on a new dataset. It will be useful to explicitly state the contributions of this work.*
>
> **R2:** This paper is an applications paper and focuses on applying neural operator to the practical of fatigue modeling. We show that using operator networks, the results can be further improved and predictions can be made very quickly, which results in very fast crack growth simulation. ICLR encourages papers showing the application to complex real-life problems and we aim to target that with this paper.
>
> ---
>
> **C3:** *Limited complexity of the dataset. The visualizations suggest that the problem involves predicting a single 1D variable that is smoothly varying. More details can be provided on the complexity of tasks considered, and a possible categorization of data samples based on the level of complexity.*
>
> **R3:** The reviewer has misunderstood the problem. We are not predicting a 1D variable that is smoothly varying. In the visualizations, we are fixing the other dimensions (that represent the plate geometry and crack shape), and only looking at the SIFs with respect to $\phi$. In reality SIFs are the function of geometry as well as the crack shape and it is not smooth in those dimensions.

---

> > ### Comment · Reviewer_J9iU · 2024-11-25
> >
> > I have read the other reviews and the author responses. As also mentioned in other reviews, this work lacks technical novelty than a simple application of existing methods to a new problem. Also, the results in response to comment C1 show that ANN (used in previous works for this problem) is almost as good as neural operator methods, limiting the contribution of this work.

---

### Meta-Review · Area_Chair_j6EZ · 2024-12-16

**Metareview:**

The paper explores the use of neural operator networks (eg. DeepONet, POD-DeepONet, FNO) for predicting Stress Intensity Factors (SIFs) in fatigue modeling. By reformulating SIF computation as an operator learning problem, the authors demonstrate significant computational speedups and higher accuracy compared to more traditional methods. The proposed approach integrates effectively into crack growth simulations and shows potential applications in assessing failure probabilities for small aircraft.

While the application is relevant and the results are promising, the paper has several limitations. The primary weakness is the lack of methodological novelty, as it applies existing neural operator frameworks without proposing innovations. Baseline comparisons with other (more standard) ML methods were missing in the original submission and the panel of reviewer estimated that they remained insufficiently addressed. The dataset is restricted to simple geometries and loading conditions, limiting insights into the method’s generalization to complex/realistic scenarios. It was also noted that there was no validation with real-world experimental data, particularly for failure probability predictions, which diminishes the practical impact of the work.

The panel recommends rejection of the paper. While the application is interesting and addresses a meaningful problem, the submission lacks sufficient innovation in methodology, rigor in baseline comparisons, and robustness in demonstrating real-world applicability. We encourage the authors to expand the scope of their work by exploring methodological enhancements, providing more comprehensive baseline comparisons, and addressing practical challenges such as generalization and uncertainty quantification. This could make the work stronger for submission to a specialized venue or a future iteration of ICLR.

**Additional Comments On Reviewer Discussion:**

During the rebuttal period, reviewers raised concerns about the lack of baseline comparisons, unclear methodological novelty, limited dataset complexity, and missing details about model architectures and hyperparameters. The authors addressed some issues by adding comparisons with ML methods & clarifying the practical relevance of their work. They also committed to providing additional methodological details in the appendix and explained the limitations of validating real-world failure probabilities due to resource constraints. However, the responses did not fully resolve the core weaknesses, including the lack of innovation, insufficient generalization testing, and the absence of experimental validation.

---

### Decision · Program_Chairs · 2025-01-22

Reject